# Exocytosis of the silicified cell wall of diatoms involves extensive membrane disintegration

Diede de Haan[1], Lior Aram [1], Hadas Peled-Zehavi [1], Yoseph Addadi [2], Oz Ben-Joseph [1], Ron Rotkopf [2], Nadav Elad [3], Katya Rechav[3] & Assaf Gal [1] ✉

Diatoms are unicellular algae characterized by silica cell walls. These silica elements are known to be formed intracellularly in membrane-bound silica deposition vesicles and exocytosed after completion. How diatoms maintain membrane homeostasis during the exocytosis of these large and rigid silica elements remains unknown. Here we study the membrane dynamics during cell wall formation and exocytosis in two model diatom species, using live-cell confocal microscopy, transmission electron microscopy and cryo-electron tomography. Our results show that during its formation, the mineral phase is in tight association with the silica deposition vesicle membranes, which form a precise mold of the delicate geometrical patterns. We find that during exocytosis, the distal silica deposition vesicle membrane and the plasma membrane gradually detach from the mineral and disintegrate in the extracellular space, without any noticeable endocytic retrieval or extracellular repurposing. We demonstrate that within the cell, the proximal silica deposition vesicle membrane becomes the new barrier between the cell and its environment, and assumes the role of a new plasma membrane. These results provide direct structural observations of diatom silica exocytosis, and point to an extraordinary mechanism in which membrane homeostasis is maintained by discarding, rather than recycling, significant membrane patches.

Diatoms are a diverse group of unicellular algae characterized by silica cell walls with intricate, species-specific shapes and hierarchical pore patterns[1]. Despite immense morphological diversity between species, most diatom cell walls have a conserved layout of two similarly shaped silica 'shells' that partially overlap, like a petri-dish. Each 'shell' in itself consists of a valve and a series of girdle bands. The valves usually define the shape of the cell, are richly ornamented, and contain hierarchical pore patterns. The girdle bands form partially overlapping rings that surround the sidewalls of the cells.

Diatom cell wall formation is under biological control and linked to the cell cycle[2–5]. Each daughter cell inherits one half of the parental cell wall and forms a second valve directly after cell division. New girdle bands are formed and appended to the new valve during the growth of the cell (Fig. S1). Silicification is usually an intracellular process, taking place inside a membrane-bound organelle, the silica deposition vesicle (SDV)[6–8]. SDVs are thin and elongated organelles, positioned under the plasma membrane. The cell regulates the chemical environment inside the SDV, thus exercising tight control over the silicification process, resulting in highly specific and reproducible cell wall architectures[9–13]. After intracellular maturation, the silica elements are exocytosed to cover the cell surface[6,14]. Such exocytosis events are exceptional in cell biology since the content of the SDV is an enormous solid structure that needs to be secreted without damaging the integrity of the cell.

In classical exocytosis pathways, the membrane of an intracellular vesicle fuses with the plasma membrane to deliver its content into the

---

[1]Department of Plant and Environmental Sciences, Weizmann Institute of Science, Rehovot, Israel. [2]Life Sciences Core Facilities, Weizmann Institute of Science, Rehovot, Israel. [3]Department of Chemical Research Support, Weizmann Institute of Science, Rehovot, Israel. ✉e-mail: assaf.gal@weizmann.ac.il

extracellular space. Homeostasis of the plasma membrane can be maintained through a transient fusion that prevents the secreting vesicle from integrating with the plasma membrane, or by offsetting the added membrane through compensatory endocytosis[15]. However, due to their large size and rigidity, the secretion of diatom cell walls inflicts a huge challenge to the organism. The new valve covers as much as half of the total cell surface (Fig. S1), and thus the surface area of the SDV membrane is similar to the entire plasma membrane. Therefore, fusion with the SDV membrane would lead to an almost instantaneous doubling of the plasma membrane area.

While former studies have brought forward several hypotheses for silica cell wall exocytosis in diatoms[14,16–23] (Fig. S2), significant experimental challenges have hindered further progress. On the one hand, traditional sample preparation for ultrastructural studies with electron microscopy causes artefacts like membrane shrinking and structure deformation[24]. On the other hand, the advantages of light microscopy in studying the dynamic aspects of cell wall formation in living cells are limited by the low spatial resolution and scarcity of molecular biology tools for diatoms[25,26]. Due to these limitations, in situ observations of the SDV in the cellular context are sparse, and direct evidence for the nature of the silicification and exocytosis process is missing.

In this study, we investigated membrane dynamics during valve formation and exocytosis in two model diatom species, *Stephanopyxis turris* and *Thalassiosira pseudonana*, using live-cell confocal fluorescence microscopy, transmission electron microscopy (TEM) and cryo-electron tomography (cryo-ET). The relatively large *S. turris* cells have easily discernible silica structures, allowing detailed observations of individual valves and the dynamic development of their architectural features using fluorescence microscopy[27,28]. In addition, using cryo-ET we obtain high-resolution 3D data of the SDV in *T. pseudonana*, at native-like conditions[29,30]. Our results indicate that valve exocytosis in both species involves disintegration of the distal membranes, accompanied by the repurposing of the proximal SDV membrane into a new plasma membrane.

## Results

### Morphology of *S. turris* and *T. pseudonana*

We investigated valve exocytosis in diatoms by studying two species, *S. turris* and *T. pseudonana* (Fig. 1). The valves of *S. turris* are capsule-shaped and have a two-layered structure. The proximal layer is thin, perforated by nano-sized pores, and on the distal side overlain by a more elevated layer that forms large polygons (Fig. 1a', a''). The top of the polygonal layer is flattened, forming a 'T'-shape in cross-section (Fig. 1a'''). *S. turris* cells form chains that are linked through tubular linking extensions of silica that extend from the apex of the valves (Fig. 1a, arrow). *T. pseudonana* cells are shaped as a barrel and are much smaller, the valves are discs with a diameter of about 5 μm (Fig. 1b). Small pores perforate the silica layer that spans the area between radial ribs, and larger tubular pores, called fultoportulae, decorate the rim of the valve[7] (Fig. 1b', b''). Figure 1e and f show dividing cells during, and shortly after, valve formation. Newly formed valves are stained by PDMPO, a fluorescent dye that is incorporated into silica during the process of biological mineralization[31]. Comparing the cell size to the size of mature valves shortly before exocytosis illustrates the enormous challenge involved in exocytosis of these rigid silica cell walls (Fig. 1g, h).

### Live-cell imaging of membrane and silica dynamics in *S. turris*

We studied membrane dynamics in *S. turris* using time-lapse confocal microscopy of living cells (Fig. 2). Synchronized cultures were labeled with PDMPO to track silica formation (Fig. S3), and with FM4-64 to stain the plasma membrane. FM4-64 is an amphiphilic fluorescent dye that integrates into the plasma membrane[32]. Unlike similar dyes, such as FM1-43 that is rapidly internalized in a different diatom[33], FM4-64 labels only the plasma membrane and does not passively infiltrate into

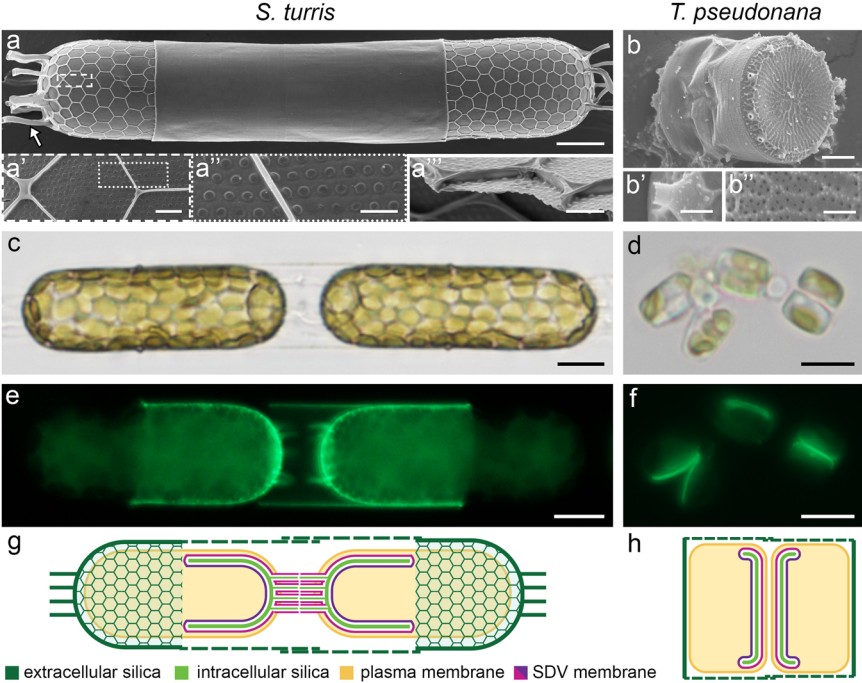

**Fig. 1 | Cell architecture of *S. turris* (a, c, e, g) and *T. pseudonana* (b, d, f, h).** **a**, **b** SEM images representing at least two samples. Details of the silica valve structures are shown at higher magnifications in the sub-panels. **c**, **d** Bright field light microscope images of two *S. turris* cells in a chain connected through linking extensions and several *T. pseudonana* cells at different stages during the cell cycle. **e**, **f** PDMPO fluorescence image of two different *S. turris* daughter cells during valve formation, and the same *T. pseudonana* cells as in (**d**) with new valves fluorescently stained by PDMPO. The results of four synchronization experiments are shown in Fig. S3. **g**, **h** Schematic of *S. turris* and *T. pseudonana* depicting the situation shortly before exocytosis of intracellularly formed valves. Scale bars: 10 μm (**a**, **c**, **e**), 5 μm (**d**, **f**), 1 μm (**a'**, **a'''**, **b**) 500 nm (**a''**), 100 nm (**b'**, **b''**).

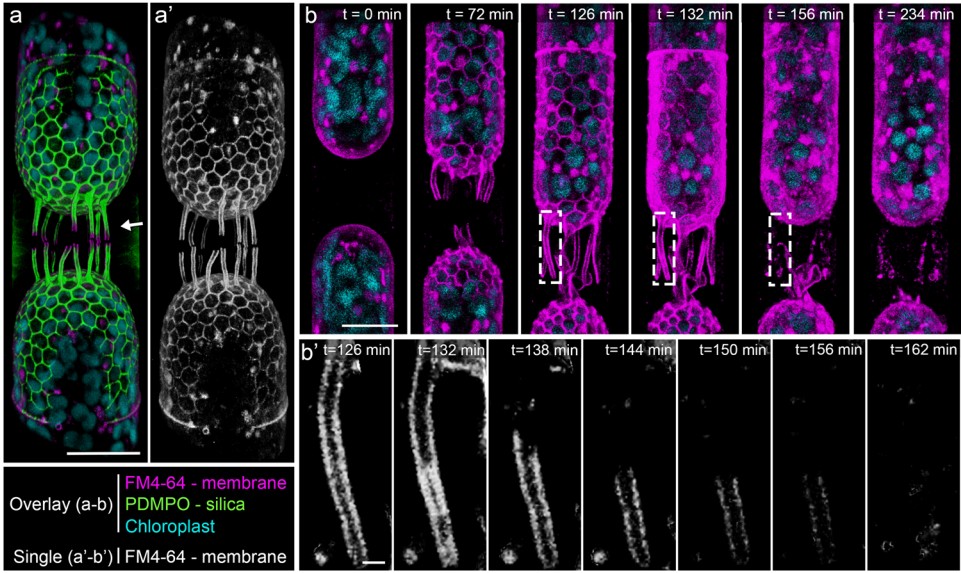

**Fig. 2 | 3D reconstructions of time-lapse confocal fluorescence images showing coordinated silica and membrane dynamics during valve formation in *S. turris*.** **a**, **b** Overlay images of fluorescence channels. **a'**, **b'** Single channel FM4-64 fluorescence images. **a**–**a'** Daughter cells during valve formation labeled with both FM4-64 and PDMPO. Membranes outline the growing polygons and lead the growth of linking extensions (arrow). **b** Snapshots from a time-lapse of daughter cells going through valve formation and exocytosis and labeled only with FM4-64. Elapsed time since start of imaging is indicated in the image. **b'** Cropped and magnified view of a single linking extension of the cell in **b** (indicated with boxes). Onset of exocytosis of the valve can be seen from *t* = 132, after which the membrane remnants around the linking extension are no longer connected to the cell and gradually disintegrate. Scale bars: 20 µm (**a**, **a'**), 10 µm (**b**), and 1 µm (**b'**).

*S. turris* cells (Fig. S4). During silicification, the PDMPO and FM4-64 signals are almost co-localized down to the resolution of the confocal microscope (Fig. 2a). This demonstrates a very close proximity between the plasma membrane and the growing silica structures within the SDV, namely that the plasma membrane is lining the SDV shape. Nevertheless, the growing tips of the linking extensions are stained only with the membrane dye, indicating that SDV elongation precedes the silicification of its most marginal parts (Fig. 2a, arrow).

We recorded a time-lapse of the membrane dynamics in *S. turris* cells going through the entire process of valve formation and exocytosis (Fig. 2b, Movie S1). Silicification starts at the cell apex and advances radially. Progress of valve formation is visualized by the fluorescent plasma membrane outlining the growing polygonal silica layer and linking extensions (Fig. 2b, *t* = 0 to *t* = 126). The onset of exocytosis, namely the loosening of the plasma membrane−SDV−silica complex is evident as the labeled plasma membrane no longer sharply outlines the polygons (Fig. 2b, *t* = 132). At the same time point, we observe an enhancement of the fluorescent signal (Fig. S5), likely due to fusion between the plasma and SDV membranes that exposes the SDV lumen to the surrounding media, from which free FM4-64 can infiltrate and stain the SDV membrane additionally to the plasma membrane. Nevertheless, the limited resolution of fluorescence microscopy does not allow to spatially resolve the interplay between SDV and plasma membranes before and after exocytosis (Fig. S6).

The linking extensions of *S. turris* present a unique opportunity to investigate the exocytosis process as the old plasma membrane surrounds these long structures before exocytosis, but after exocytosis, the new plasma membrane is only at their base. The FM4-64 signal around the linking extensions is continuous during their growth, but after the extensions reached their full size the fluorescent signal changes to interrupted patches that gradually disappear (Fig. 2b, *t* = 156 and *t* = 234). This is in accordance with a previously reported membrane fluorescence study of *Coscinodiscus wailesii*[33]. In some cases, the membranes surrounding a linking extension are clearly no longer connected to the cell body (Fig. 2b'). This labeling pattern, in which patches of stained membranes are

disconnected from the cell, is very different from the expectation for a classical exocytosis process where stained membranes should be withdrawn and recycled inside the cell. It is important to note that FM4-64 is present in the medium throughout the experiment but only becomes fluorescent in a lipid environment[32]. Therefore, it labels only structurally intact membranes and when a membrane is disintegrated it will no longer label the malformed debris. Thus, these observations point to a scenario where major fractions of the distal membranes are completely detached from the main cell surface, gradually disintegrate, and lose their fluorescent labeling in the extracellular space between recently divided daughter cells.

### Ultrastructure of silica formation and exocytosis in *S. turris*

To observe the same process at ultrastructural resolution we prepared dividing *S. turris* cells for TEM analysis. In short, synchronized cells were vitrified using high-pressure freezing and subsequently freeze-substituted and embedded in resin. This results in an optimal combination of near-to-native state preservation of subcellular structures paired with the ability of high-throughput TEM imaging at room temperature. The TEM images of *S. turris* cells prepared according to this procedure show exceptional preservation of the cellular environment and notably the membranes and inorganic contents of the SDV (Fig. S7).

We acquired hundreds of images from 227 cells at different stages of the cell cycle, and by categorizing and sorting these images, we reconstructed a timeline of valve formation and exocytosis in *S. turris* (Fig. 3). Cells with an SDV containing a growing valve were classified as being at the stage of valve formation (*n* = 61, examples presented in Fig. 3a–b"). At these stages three lipid bilayers, the proximal and distal SDV membranes, as well as the plasma membrane, can be clearly distinguished. The distal SDV membrane is in very close proximity to the plasma membrane, with a distance of only 10−30 nm (Fig. 3a" inset). During early valve formation, the SDV extends only as far as silica deposition has advanced (Fig. 3a"). As silica precipitation proceeds radially, the SDV expands with it. After formation of the porous base layer, the polygonal layer is formed on its distal side (Fig. 3b–b", arrow). During the entire silicification process, the SDV membrane

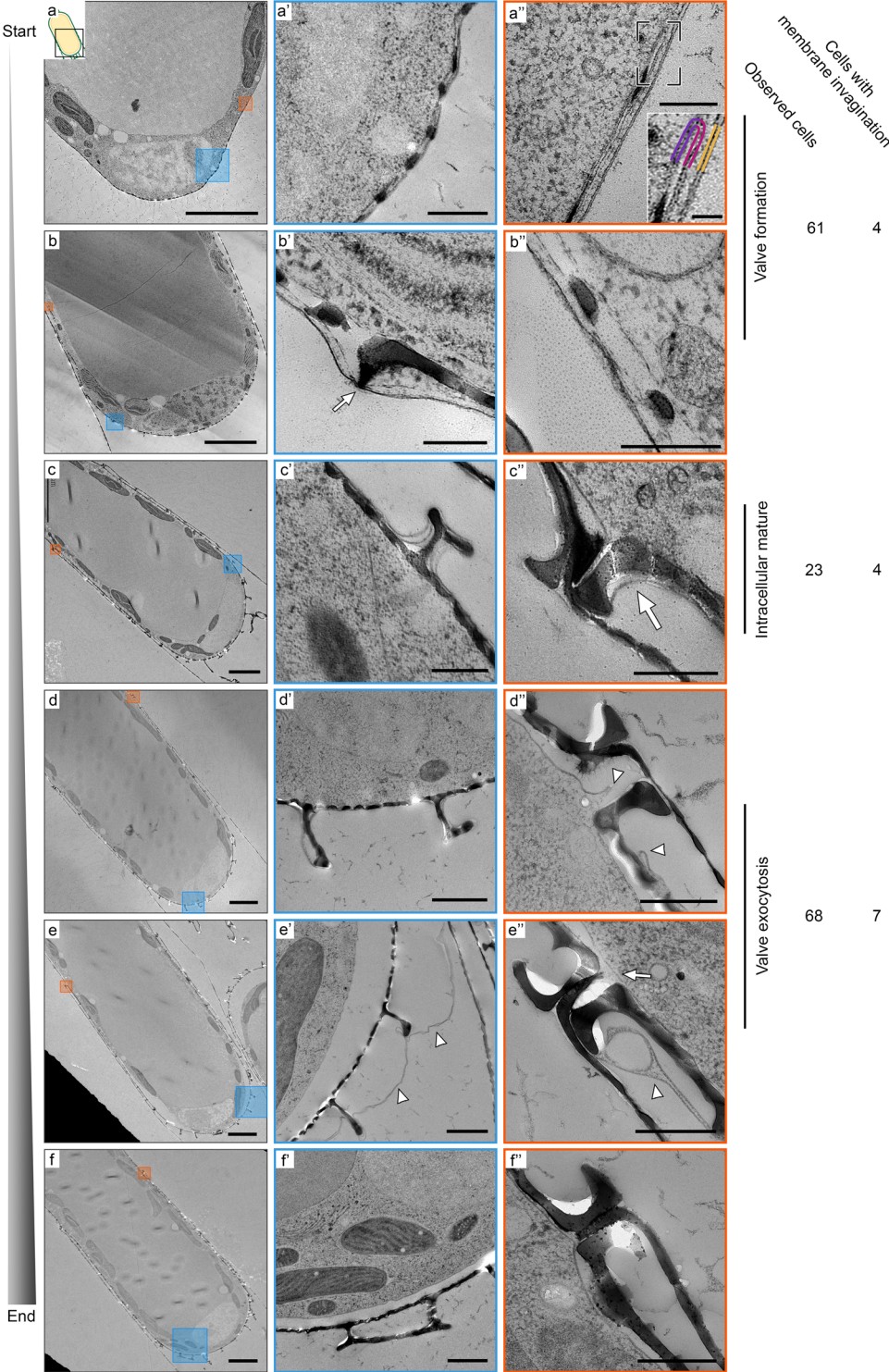

**Fig. 3 | TEM images showing sequential stages of valve formation and exocytosis in *S. turris*.** Left column shows a representative image of each stage, with higher magnification views of the boxed areas in the right columns. **a** Valve formation starts at the apex of the cell. **a'** Growing silica inside the SDV that is located directly under the plasma membrane. **a"** The SDV ends at the growing silica edge. The inset shows the bilayers of the proximal (purple) and distal (pink) SDV and the plasma membrane (yellow) in higher magnification. **b** Later during valve formation, **b'** the polygonal layer (arrow) is formed on top of the base layer, and **b"** the growing edge of the new valve reaches the rim of the parental valve. **c** Silicification of the new valve is completed while it is still fully enclosed in the SDV. Mature valves can be recognized by: **c'** the fully formed and flattened polygonal layer, and **c"**

a pronounced hook shape at the valve rim (arrow). **d** At the onset of exocytosis, **d'** membranes still surround most parts of the new valve, **d"** but they no longer form a complete enclosure (arrowheads). **e** Shortly after exocytosis, **e'** distal membranes lose their structural integrity (arrowheads), and **e"** the plasma membrane is now continuous (arrow) under the new valve. **f–f"** After completion of exocytosis (*n* = 75), there are no traces left of the distal membrane remnants. To the right of the images, a table summarizes the number of cells observed in each developmental stage and the number of cases when membrane invaginations were present (see details in Fig. S8 and main text). Scale bars represent 5 μm (**a–f**), 1 μm (**d'**, **e'**, **f'**), 500 nm (**a'**, **c'**, **c"**, **d"**, **e"**, **f"**), 200 nm (**a"**, **b'**, **b"**, **c'** inset) and 50 nm (**a"** inset).

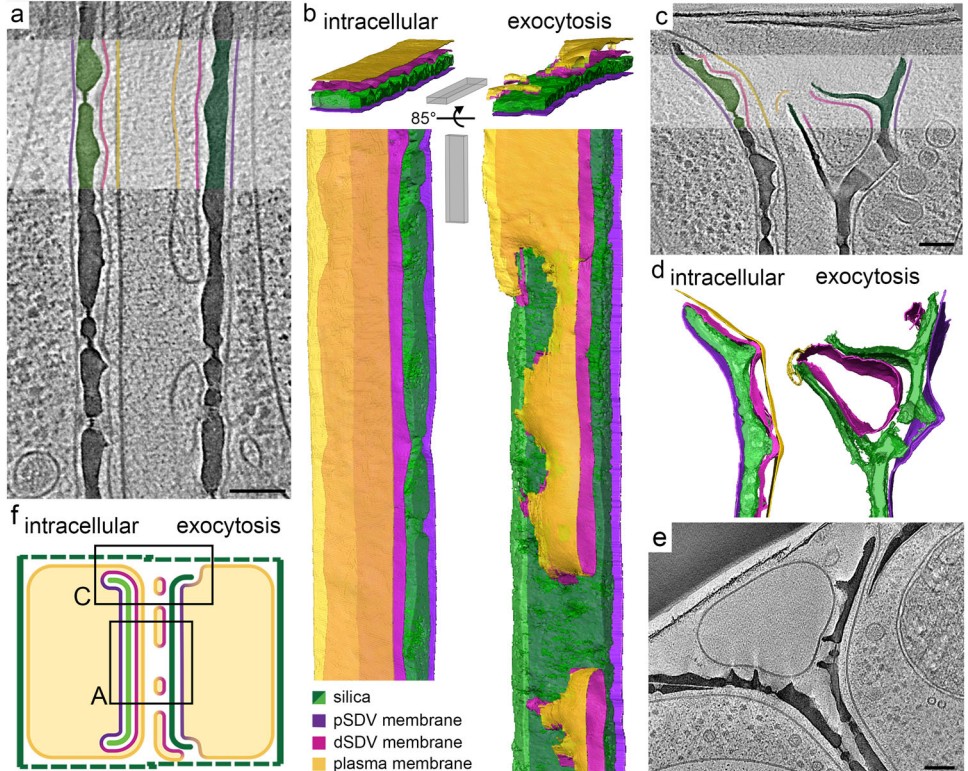

**Fig. 4 | In-cell cryo-ET shows the native-state anatomy of valve exocytosis in *T. pseudonana*. a, c, e** Slices through reconstructed 3D volumes; some features in the highlighted rectangles are artificially colored according to the color code in **b** (pSDV membrane - proximal SDV membrane, dSDV membrane - distal SDV membrane). **b, d** Three-dimensional surface rendering of segmented volumes. **f** Schematic showing the proposed valve exocytosis mechanism of diatoms. **a** and **c** show different locations on the same daughter cells, indicated in **f**. Scale bars: 100 nm, slice thickness 1.15–5.7 nm. See also Movies S2, 3.

tightly delineates the growing valve, forming a highly confined space in which the silica morphology is under control of the SDV membrane. The valve reaches its final morphology while it is still fully enclosed in the SDV, we observed 23 cells with an intact SDV containing a fully mature valve (Fig. 3c–c").

In images of 68 cells, we observed a mature valve that is surrounded by SDV membranes with some discontinuities, i.e., the SDV does not form a complete enclosure. Based on this structural information we define these cells as undergoing exocytosis. These membrane discontinuities can be very local, while the majority of the valve is still tightly enclosed by the SDV (Fig. 3d–d"). At a later stage, the membrane ultrastructure changes dramatically. The tight delineation of the distal SDV and plasma membrane around the valve loosens, and we observe deterioration of the structural integrity of the distal membranes (Fig. 3e–e"). The membrane at the proximal side of the valve is now continuous with the plasma membrane under the parental valve (Fig. 3e" arrow). At the end of the exocytosis process, the new valves can be recognized due to their position under the parental girdle bands, while no visible traces of the membranes are seen outside the cytoplasm ($n = 75$, Fig. 3f–f").

We analyzed the microscopy data of the membrane proximal to the newly formed valves in order to identify possible signs of endocytic membrane retrieval that might be associated with recycling the SDV membranes. We observed seven occasions of membrane invaginations in cells at the exocytosis stage. The size of these invaginations varies from 0.5 to 3 µm (Fig. S8). However, such invaginations were detected at all stages of valve formation and exocytosis with similar occurrence ($X^2_{(df=2, N=152)} = 2.23$, $p = 0.327$), and thus cannot be correlated to membrane recycling after exocytosis. Overall, the TEM data is in agreement with the live-cell imaging, suggesting that during valve exocytosis the distal membranes disintegrate extracellularly without

signs of retrieval, while the only visible membrane proximal to the new valve is the proximal SDV membrane.

## Native-state nanoscale architecture of silica exocytosis in *T. pseudonana*

In order to visualize the cellular organization as close as possible to the native state, we acquired cryo-electron tomography (cryo-ET) data from *T. pseudonana* cells undergoing valve exocytosis. The smaller size of these cells makes them suitable for the needed sample preparation steps for imaging with cryo-ET. In short, we vitrified synchronized *T. pseudonana* cells using plunge freezing, prepared thin lamellae using a focused ion beam and then collected electron tomograms under cryogenic conditions[30,34]. Out of the 59 pairs of daughter cells that were imaged, 15 were during or shortly after valve exocytosis, i.e., before exocytosis of the first set of girdle bands. Four of those were during the early stages of exocytosis, with the new valve only partly covered by SDV membranes (Fig. 4a–d).

Figure 4a, c shows tomograms of different locations of the same pair of slightly unsynchronized daughter cells, the left one shortly before exocytosis and the right one during exocytosis. The valve on the left side is still completely enclosed within the SDV, while the valve on the right side is already exposed to the extracellular space through discontinuities in the coverage by the distal membranes. In both cells, three lipid bilayers are discernible. In the daughter cell before exocytosis the expected arrangement of membranes is seen: the plasma membrane covers the whole cell and closely underneath the SDV membranes fully surround the valve (Fig. 4a, b, left side). However, during exocytosis the distal membranes have fused at multiple sites, forming a network of flat membrane sacs, making the membrane on the proximal side of the valve the outermost boundary of the cell (Fig. 4a, b, right side). A similar situation is observed at the valve

periphery, where unconnected membranous structures are seen at the distal side of the fultoportula (Fig. 4c, d, right side).

In eight pairs of cells that were at later stages of exocytosis we observed large membranous vesicles positioned between the recently exocytosed valves of two daughter cells and their girdle bands (Fig. 4e). In the three remaining pairs of cells with recently exocytosed valves we did not observe any vesicles or membrane remnants between the two daughter cells. However, in those three cells the parental girdle bands were not enclosing the daughter cells, pointing to a later stage of the cell cycle, by which all extracellular debris had been degraded. Notably, none of the imaged cells contained budding endocytic vesicles or a sign for the formation of a new membrane underneath the valve. Therefore, the cryo-ET data suggest that *T. pseudonana* uses the same exocytosis mechanism that we inferred from the data collected for *S. turris*, where the proximal SDV membrane is repurposed as a new plasma membrane and the distal membranes disintegrate extracellularly (Fig. 4f).

## Discussion

Extrusion of the rigid cell walls of diatoms is an exceptional exocytosis event, difficult to reconcile with known exocytosis mechanisms. Our observations, of detached membrane patches in the extracellular space of two diatom species acquired with three microscopy techniques, are incompatible with classical exocytosis mechanisms. The structural characteristics of these unconnected and loose membranes after valve formation are in sharp contrast to the SDV ultrastructure during silica growth, which forms a highly confined space that is pivotal in controlling the shape of growing silica structures[35–37]. Therefore, we suggest that exocytosis of mature valves occurs via a unique mechanism in which the old plasma membrane and distal SDV membrane are discarded in the extracellular space and the proximal SDV membrane takes the role of a new plasma membrane (Figs. 4f, S2). This scenario is in agreement with previous observations of rapid exchange of SDV proteins with the plasma membrane in live-cell studies of *T. pseudonana*[2,38].

This scenario is also supported by the fact that none of the cells in our datasets show signs of processing or recycling of the SDV membranes, or the formation of a new plasma membrane under the valve. Nevertheless, we are aware that our live-cell imaging might not have the spatial resolution to detect such events, and that electron microscopy only gives information on random snapshots in time and space. For these reasons, we conducted a statistical analysis to determine the probability that exocytosis does involve the endocytic recycling of the SDV membrane. The tested scenario was that the ~1 μm invaginations observed in *S. turris* are actually part of such endocytic retrieval process (Fig. S8). If this is the case, it will require the recycling of ~1300 such invaginations. We ran a statistical simulation (see Supporting information section), assuming a 10–20 s time window for an invagination to develop and be visible in an image, and a total 30 min for valve exocytosis. The simulation also addresses the probability to 'catch' this event in a random TEM thin slice. Running the simulation for 10,000 times using the permissive conditions of 10 s for an invagination, showed that such invaginations should be detected on average 15 times out of 68 observations (corresponding to the 68 actual observations, Fig. 3). Moreover, in more than 99.9% of the cases, the simulated scenario included the detection of more than 7 such events when 68 observations were made (corresponding to the 7 observed cases, Fig. S8). Therefore, we can rule out the option of internalization of these membranes through compensatory endocytosis with a probability of $p < 0.001$, according to the permutation test.

The proposed scenario suggests that upon exocytosis, the proximal SDV membrane becomes the new plasma membrane. As each of the SDV and plasma membranes carry out specialized, different tasks, it is likely that both require unique lipid and protein compositions, which would need to be adjusted after fusion. Indeed, it has been shown that *T. pseudonana* cells with SDVs have a slightly altered lipid composition, compared to cells that do not contain SDVs, and several proteins have been shown to be specifically associated with the SDV membrane[2,39–41]. Possible alterations to the proximal SDV membrane to restore plasma membrane specific lipid composition can be mediated by so-called lipid transfer proteins, lipid carriers that transfer specific lipids between donor and acceptor membranes[42]. The finding that the distal membranes disintegrate extracellularly is surprising, as it seems wasteful to discard such a large amount of membranes each cell cycle. The low affinity of FM4-64 to degrading debris precludes the ability to follow the fate of these membranes[32]. Nevertheless, it is important to note that the girdle bands form a quasi-enclosed compartment around the newly formed valves, possibly slowing down diffusion away from the cell and facilitating the uptake of the membrane debris for cellular use after disintegration. This can conceivably minimize the energetic costs of such process.

The exocytosis of large content was also investigated in other organisms, demonstrating mechanisms that differ from the classical secretory pathway of numerous small vesicles that fuse with the membrane[43]. One alternative is the exocytosis of giant vesicles in the *Drosophila* salivary glands that squeeze out their viscous cargo through a pore by crumpling of the vesicular membrane, followed by membrane recycling[44]. Another alternative, the acrosome reaction, surprisingly shares similarities with our proposal for diatom valve exocytosis. In this process, the plasma membrane and distal acrosomal membrane fuse at several locations forming vesicles that are dispersed in the environment while the proximal acrosomal membrane remains intact and becomes the new boundary that separates the spermatozoa from the environment[45,46].

To conclude, this work proposes a unique mechanism for the exocytosis of diatom silica, which involves two remarkable events. First, the repurposing of an organelle membrane into a plasma membrane and second, large-scale disintegration of membranes. This mechanism is shared by two model diatom species, indicating that this might be a general mechanism. With the toolbox for genetic research in diatoms growing, it will soon be possible to investigate the protein machinery that is involved in the regulation of this event, and its relation to classical exocytosis.

## Methods

### Cell cultures

*S. turris* was isolated from the North Sea in 2004 and provided by the group of Prof. Eike Brunner, TU Dresden. Diatom cultures were maintained in natural Mediterranean seawater that was filtered, its salinity corrected to 3.5% and supplemented with f/2 nutrient recipe (Sigma Aldrich), and for *Thalassiosira pseudonana* (CCMP1335) supplemented with 330 μM silicic acid (Sigma Aldrich). Cultures were maintained at 18 °C under 16/8 h light/dark cycles.

### Cell cycle synchronization

For synchronization of the cell cycle, 5 ml of a mature *S. turris* cell culture was used to start a new 50 ml culture. After 48 h of growth under the normal 16 L/8 D cycle, the culture was placed in darkness for 20 h. After 20 h of darkness, the culture was again exposed to light. *T. pseudonana* synchronization was done using Si starvation, as previously described[7]. To maintain the culture in an exponential growth phase they were grown under 12/12 h light/dark cycles and diluted (1:10) into fresh medium every other day during the week before culture synchronization. To induce Si starvation, 100 ml aliquots of culture were centrifuged at 3000 × g for 10 min and re-suspended in Si-free artificial seawater or filtered seawater; this step was repeated three times. The cultures (~0.5 million cells/ml), were then maintained in dark for 12 h under agitation in a Si free medium to arrest the cell cycle. Then cultures were transferred to continuous light for an additional 4 h of Si starvation. At the end of the Si starvation period, cells were

concentrated to about 10 million cells/ml and Si was replenished to 330 μM. To track the formation of new silica, PDMPO [2-(4-pyridyl)−5-((4-(2-dimethylaminoethyl-amino-carbamoyl)methoxy)-phenyl) oxazole] (ThermoFisher Scientific, USA) was added. PDMPO fluorescence was monitored by imaging the cultures with an epifluorescence microscope (Nikon Eclipse Ni-U, ex: 365 em: 525). Highest amount of dividing *S. turris* and *T. pseudonana* cell were counted after 9 and 3 h, respectively.

### Sample preparation for SEM
Cells were prepared for SEM using critical point drying (CPD). Cells were fixed in a solution of 2% Glutaraldehyde and 4% Paraformaldehyde in artificial seawater for 1 h at room temperature while shaking. After three washes with deionized water (Milli-Q® IQ 7003 Ultrapure Lab Water System, Merck), the cells were dehydrated by washing in a graded series of ethanol. The final wash was done in 100% anhydrous ethanol overnight. The dehydrated samples were then dried in a critical point dryer using liquid $CO_2$ as transitional fluid. Dried cells were placed onto a conductive carbon tape on an aluminum stub.

### SEM imaging
Samples were sputter-coated with 2.5 nm (*T. pseudonana*) or 4 nm (*S. turris*) iridium (Safematic) and imaged with an Ultra 55 FEG scanning electron microscope (Zeiss, Germany), using 3–5 kV, aperture size 20–30 μm and a working distance of about 3 mm.

### Live-cell imaging
For single-cell time-lapse imaging, a culture was synchronized and stained with PDMPO (330 μM). About 8 h after the end of light starvation, when most cells had gone through cytokinesis, an aliquot of 100 μl was taken and FM4-64 (ThermoFisher Scientific, USA) was added to a final concentration of 4–8 μM to stain the membranes. After adding the membrane dye, a 20 μl drop was mounted on a microscope slide and covered with a glass coverslip, using dental wax as spacer. The samples were visualized using a Leica TCS SP8-STED confocal microscope equipped with a HCS PL APO 86×/1.20 W motCORR objective. FM4-64 and chlorophyll autofluorescence were acquired by white-light laser using 550 nm laser line (6% laser power) and 650 nm laser line (7% laser power), respectively. PDMPO fluorescence was acquired using a 405 nm laser (5% laser power). HyD-SMD detectors were used for PDMPO and FM4-64 with emission collection width set to 474–530 and 608–640 nm, respectively. Chlorophyll autofluorescence emission was collected using a HyD detector with 741–779 nm detection width and the transmission channel was detected with a PMT detector. The cells were imaged for 2–4 h at intervals of 3–60 min. Images were analyzed using Leica Application Suite X v3.5.7. In total, time lapses from over 50 cells were collected over the entire period.

### Sample preparation for TEM
*S. turris* cells were cryo-fixed, using high pressure freezing (HPF), followed by freeze substitution (FS), according to previously published protocols[29,47]. Synchronized cells were collected on a 5 μm filter membrane and transferred to an Eppendorf tube using 200 μl of seawater. After letting the cells sediment for 10 min, 2 μl aliquots were pipetted into aluminium discs (Wohlwend GmbH, Sennwald, Switzerland) and directly loaded into a Leica ICE high pressure freezing machine (Leica Microsystems GmbH, Wetzlar, Germany). Concentrated diatom samples were vitrified in liquid nitrogen (−192 °C) at 210 MPa (2048 bar). Vitrified samples were stored in liquid nitrogen until freeze-substitution in an EM ASF2 (Leica Microsystems GmbH, Wetzlar, Germany). Vitrified water was substituted with an organic solvent, 100% anhydrous acetone, at (−90 °C). Then, acetone was supplemented with chemical fixatives (0.2% uranyl acetate and 0.2% osmium tetroxide) to enhance cross-linking and contrast of cellular

structures. The samples were immersed in the solution for 48 h at −90 °C and then allowed to gradually warm for 24 h to −20 °C, and then in one hour to 0 °C. After three washes in acetone, the acetone was replaced with Epon (Agar Scientific Ltd, Stansted, U.K.) using gradient concentration mixtures (10%, 20%, 30%, 40%, 60%, 80%, 100% Epon in acetone), twice a day at room temperature. The sample in 100% Epon was hardened at 70 °C for 72 h. Ultrathin sections of 70 nm were sliced using an ultra-microtome (Ultracut UCT, Leica Microsystems GmbH, Wetzlar, Germany) equipped with a diamond knife (Ultra 45°, Diatome Ltd, Nidau, Switzerland). The sections were picked up onto copper TEM grids, coated with carbon film. The sections were then post-stained by placing them onto a drop of lead citrate solution. After three minutes of staining, the grids were washed three times in drops of water and then dried by blotting onto whatman filter paper. In total, we prepared 16 frozen samples that were further processed using freeze substitution during three different cycles, and subsequently imaged hundreds of cells.

### TEM imaging
The TEM samples were imaged with a Tecnai Spirit TEM (FEI, Eindhoven, Netherlands) operated at 120 kV and equipped with Gatan Oneview 4 k × 4 k camera (Gatan Inc., Pleasanton, U.S.A.).

### Plunge freezing
Synchronized *T. pseudonana* cells were vitrified by plunge-freezing on glow discharged 200 mesh copper R2/1 holey carbon film grids (Quantifoil Micro Tools GmbH, Grossloebichau, Germany). In a Leica EM GP (Leica Microsystems GmbH, Wetzlar, Germany), 1 μl of artificial seawater was pipetted on the copper side in order to enhance media flow to the blotting paper and 4 μl of cell suspension at 7–13 × 10⁶ cells/ ml was pipetted on the carbon side. The grids were blotted for 6 seconds from the back side of the grid before they were plunged into a liquid ethane bath cooled by liquid nitrogen.

### Cryo-FIB milling
Vitrified cells were milled to thin lamellae with the Zeiss Crossbeam 550 FIB/SEM dual beam microscope (Zeiss, Germany). The grids were coated with organometallic platinum by an in situ Gas Injection System. The lamellae were milled at a 12° tilt relative to the grid plane with the rough milling (to 1 μm thickness) involving two steps using the gallium ion beam at 30 kV and a current of 300 pA and 100 pA. After rough milling all lamellae, they were thinned to 200 nm at a current of 50 pA.

### Cryo-electron tomography and volume rendering
Cryo-electron tomography data were collected from 59 pairs of cells. The tilt series were acquired using a Titan Krios G3i TEM (Thermo-Fisher Scientific, Eindhoven, The Netherlands), operating at 300 kV. Tilt series were recorded on a K3 direct detector (Gatan Inc., Pleasanton, U.S.A) installed behind a BioQuantum energy filter (Gatan Inc., Pleasanton, U.S.A), using a slit of 20 eV. All tilt series were recorded in counting mode at a nominal magnification of 33,000×, corresponding to a physical pixel size of 0.26 nm, using the dose-symmetric scheme starting from the lamella pre-tilt of −12° and with 2° increments[48]. Tilt series appearing in Fig. 4a, c were taken at 3 μm defocus and using a Volta Phase Plate inserted. The tilt series range was between −60° and 50°. Tilt series appearing in Fig. 4e was taken at 7 μm defocus, an objective aperture of 100 μm inserted, and −66° to 48° tilt range. Tilt series were acquired using an automated low dose procedure implemented in SerialEM v3.8 with a total dose set to -100e-/Å²[49]. The tomograms were reconstructed using IMOD software 4.9.12[50]. Amira software v2021.2 was used for segmentation (Thermo-Fisher Scientific, Eindhoven, The Netherlands). Membranes were segmented using the membrane enhancement filter module and manually refinement[51].

## Statistical simulations and analyses

Our data indicate that diatoms use a surprising, non-canonical mechanism to exocytose their silica valves. To consider an alternative scenario, in which diatoms use a known canonical mechanism of valve exocytosis, we performed a statistical simulation that will inform whether such a scenario is likely. In canonical exocytosis, the membrane of the secreting vesicle fuses completely with the plasma membrane, and membrane homeostasis is maintained by balancing out the added membrane by compensatory endocytosis. In the putative scenario that we want to investigate the SDV membrane is recycled by compensatory endocytosis of vesicles of constant size. In order to simulate such a scenario we had to estimate how many vesicles have to be endocytosed to re-internalize the SDV membrane upon valve exocytosis and how many of such vesicles should have appeared in our dataset.

### Estimation of the number of putative vesicles that are generated during SDV membrane recycling

**Calculation.** From basic geometrical properties of the diatom cell the surface area of both proximal and distal SDV membranes (excluding the membranes around linking extensions) ≈ surface area of the entire cell that can be simplified into a capsule (Fig. S9). We therefore calculated the area of such a capsule:

A capsule is a cylinder with hemispheres on either end. The surface area of a capsule is defined as the sum of the surface area of the cylinder in the center and the combined surface area of the two hemispheres. To calculate the capsule surface area one needs the radius (r) of the hemispheres and the length of the central cylinder (a). *S. turris* cells are 20–25 μm in diameter, thus in the calculation we chose a radius of 11 μm. The length of the cylinder (a) is equal to the total length of the cell minus the radii of the two hemispheres (60 μm – 11 μm – 11 μm = 38 μm). Therefore the surface area (SA) of the capsule (and the SDV membrane) is:

$$SA_{capsule} = 2\pi r(2r + a) \approx 4000\ \mu m^2$$

**Estimation of membrane surface area of endocytic vesicles.** In our TEM data of *S. turris* we observed membrane invaginations with diameters ranging from 0.5 to 1 μm. The surface area of such invaginations can be calculated by approximating their geometry to that of a sphere: $SA_{sphere} = 4\pi r^2$

$$r = 0.25\ to\ 0.5\ \mu m$$

$$SA_{sphere} = 0.78\ \mu m^2\ to\ 3.14\ \mu m^2$$

**Estimating the number of vesicles to be endocytosed.** Based on the previous calculations, it follows that the number of endocytic vesicles required to recycle the entire SDV membrane is:

$$SA_{SDV}/SA_{vesicle} = Number\ of\ endocytic\ vesicles(1\ \mu m\ in\ diameter)$$
$$= 4000/3.14 = 1274\ vesicles$$

**Duration of an endocytosis event for a vesicle of 1 μm.** Studies of time resolved endocytosis indicate that a single endocytosis event of a vesicle with a diameter of 1 μm should take at least 10 s[49–55].

**Total duration of the exocytosis/endocytosis process.** Based on our data, we surmise that the recycling process should be finished in 30 min.

**Calculating the chance of catching an endocytic vesicle in a TEM section.** In addition to the temporal factors in sections 4 and 5 that influence the chances of observing an endocytic vesicle, there is a spatial factor which is the chance of detecting such a vesicle in a random TEM slice. Since the cells are capsule-shaped, lying roughly horizontally in the embedded samples, we can estimate the spatial chance of detecting a vesicle with diameter x in a random slice going through the round cross section of a cell with diameter y as (Fig. S9): $\frac{x}{y} = \frac{1}{22}$ for a vesicle of 1 μm in a TEM section of an *S. turris* cell of 22 μm.

**Permutation simulation.** Given the four parameters identified in the previous sections we build a simulated scenario (Using R, v. 4.1.2) in which there are 1274 events (endocytic vesicle) that last for 10 seconds within a 30 min time frame, and the simulation is picking random snapshots (TEM slices) in which the chance to visualize an event (if indeed happening) is 0.045 (1/22). Since our TEM dataset includes seven possible endocytic vesicles in the 68 *S. turris* cells that were undergoing exocytosis, we simulated what are the chances that seven such events will be observed given 68 observations. The simulation shows that in >99.5% of the cases more than seven endocytic events should be detected (Fig. S10A).

## Reporting summary

Further information on research design is available in the Nature Portfolio Reporting Summary linked to this article.

## Data availability

All relevant data supporting the key findings of this study are available within the article and its Supplementary Information files or from the corresponding author upon reasonable request. Source data are provided with this paper and on Dryad, https://doi.org/10.5061/dryad.gxd2547n3. Source data are provided with this paper.

## Code availability

The code used in this study is available through the following link: https://zenodo.org/record/7339127#.Y58SgXZBwuU.

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

## Acknowledgements

We thank Anne Jantschke, Nathalie Pytlik, and Eike Brunner from TU Dresden for providing cultures. We thank Simon Michaeli from the Volcani Institute (ARO) for his contribution in finding the right live-cell imaging platform. This project has received funding from the European Research Council (ERC) under the European Union's Horizon 2020 research and innovation programme (grant agreement No. 848339). Live-cell imaging was conducted at the de Picciotto Cancer Cell Observatory In Memory of Wolfgang and Ruth Lesser. The work was supported by the Irving and Cherna Moskowitz Center for Nano and Bio-Nano Imaging at the Weizmann Institute of Science D.dH. was supported by the Sustainability and Energy Research Initiative (SAERI) of Weizmann Institute of Science.

## Author contributions

D.dH., performed experiments, analyzed data, drafted and edited the manuscript. L.A. performed cryoET experiments and data analysis. H.P.Z. performed confocal microscopy experiments. Y.A. contributed to the experimental design of the confocal microscopy experiments. O.B. assisted with room temperature TEM sample preparation. R.R. conducted the statistical analyses. N.E. contributed to the cryoET experiments and data analysis. K.R. assisted with FIB lamellae preparation. A.G. provided supervision, funding acquisition and revised drafts.

## Competing interests

The authors declare no competing interests.
