## [Peer Review File · Nature Communications]

REVIEWER COMMENTS

Reviewer #1 (Remarks to the Author):

The paper by de Haan et al. reports on the membrane dynamics and exocytosis of the silica-based cell wall in diatoms. The mechanisms underlying the intracellular production of an entire mineral-based cell wall and then exocytosis of a rigid structure has been of interest to diatom biologists for decades and various models of exocytosis have been proposed. de Haan uses state-of-the-art microscopy on two diatom species to provide empirical evidence supporting the model that proposes fusion of the proximal SDV membrane with the plasma membrane and then exocytosis of the cell wall. They propose that the proximal SDV membrane is then remodeled and becomes the new plasma membrane of the daughter cell and distal SDV membrane undergoes disintegration. As the authors state, the novelty of this study is the proposition that half of the silicalemma becomes a plasma membrane and the other half is disintegrated and possibly lost.

I am not a structural biologist so I will leave a review of the methodology and validity of the conclusions to others. I will thus base my review on the assumption that the authors' interpretation of the results is robust. All of the models of how diatoms exocytose a fully formed cell wall require some level of fusion between the silicalemma and plasma membrane. What differs among the models is the manner in which membrane homeostasis is maintained. This ranges from membrane recycling through vesicles to repurposing the membrane as the protective organic layer surrounding the frustule. In this study, de Haan et al. proposes a modified mechanism that involves fusion of the proximal SDV membrane which later becomes the new plasma membrane, followed by exocytosis of the cell wall and subsequent disintegration of the remaining silicalemma. Kotzsch et al. 2017 also provides evidence that the proximal SDV membrane fuses with the plasma membrane and although their data hints at distal silicalemma recycling via endocytotic vesicles, they acknowledge the uncertainty of this latter conclusion based on their limited data. So the novelty and significance de Haan et al. can provide is insight into the fate of the distal silicalemma.

Remodeling of the silicalemma as a plasma membrane is an intriguing idea that opens up a whole slew of question. Is this even possible? How would the cell 'swap out' silicalemma-associated proteins with plasma-membrane associated ones? What is known about the lipid composition of the silicalemma vs the plasma membrane? This is not a criticism but rather an acknowledgement that this work stimulates a number of testable hypotheses (which all good science should do) that would hopefully get us closer to understanding the mechanism of cell division in diatoms. However, I do think this could have been discussed more rather than just stating that 'it is likely that both require unique lipid and protein compositions, which would need to be adjusted after fusions' (Lines 275-276). Just a couple of lines about whether there is precedence for this and some citations would at least give some food for thought.

The second novel idea de Haan et al. proposes is large-scale disintegration of the distal SDV membrane in the extracellular space, essentially being loss to the environment, contrasting previous suggestions that the membrane is recycled through endocytotic mechanisms. As support for their conclusions, de Haan et al. run a simulation on the probability of detecting endocytosis. Based on these simulations, they say they would expect to detect 15 endocytotic events out of 68 observations and since they only observed 7, they conclude that membrane internalization cannot be significant. (Lines 270-272). This calculation is somewhat critical to rule out the possibility of membrane recycling as they indicate on Lines 254-258 and support their conclusion of membrane disintegration, but there are some aspects of it that are unclear. First, the surface area of the mature SDV is calculated as $\sim 4000 \mu\text{m}^2$ which they state is equivalent to the surface area of the plasma membrane. First, I don't understand why these are equivalent. One daughter cell SDV is not the same size as the entire cell. Where does $a = 38$ come from? This should be the side length (incidentally the authors should define these in the methods so those of us who have forgotten geometry don't have to look up the formula)

but they state that the cell is 60 um long. Even if I assume they have taken into account there being 2 daughter cells, I still don't see where 38 comes from. Furthermore, the authors conclude that it is the distal SDV membrane that is being disintegrated so shouldn't their simulation reflect that? If they calculate 15 out of 68 observations for a full SDV, does that mean it would be 7.5 out of 68 for half the SDV? In which case, can they confidently rule out membrane recycling if they indeed observed 7 cases of membrane invagination?

I think this is overall a very nice study with high quality and modern microscopic approaches and does provide additional support for the mechanism of cell wall exocytosis in diatoms. However, the claim in the title seems to depend heavily on a statistical simulation that, at first pass, does not seem robust and should be clarified.

Reviewer #2 (Remarks to the Author):

This manuscript describes the process of frustule formation in diatoms, specifically the exocytosis of the mature valve, a remarkable exocytosis event given the size of the newly-formed valve relative to the cell. They demonstrate that the plasma membrane and distal SDV membrane are lost, and the proximal SDV membrane becomes continuous with the plasma membrane.

Overall, the study is conducted to a very high degree. The resolution of the imaging, particularly the cryo-ET, is exceptional and provides excellent detail of the process of valve exocytosis. Overall, I find that the findings strongly support the proposed mechanism. The study is of broad interest, not only to the study of biomineralisation but also to cell biologists more generally. I only have a few technical points relating primarily to the live cell imaging that need to be addressed prior to publication.

Specific comments

1) Distribution of FM4-64. The live cell imaging of frustule formation and exocytosis used FM4-64 to stain the plasma membrane (Fig 2). Imaging took place over several hours (up to 4 hours). FM4-64 was included in the media throughout the imaging period. The 3D images presented are beautiful and demonstrate that FM4-64 is concentrated in the polygonal pattern seen for PDMPO. I initially found this quite confusing as you might expect the plasma membrane to simply form a smooth continuous layer above the developing SDV. However, the arrangement of the plasma membrane over the edge of the polygonal silica structures can be clearly seen in Fig3C, and this may explain why FM4-64 exhibits this distribution. The authors do say that this is because of 'close proximity of the plasma membrane to the SDV' (line 101) but I feel this needs more explanation.

2) Internalization of FM4-64. Previous studies using FM dyes in diatoms have shown that they are rapidly internalized e.g. FM1-43 was rapidly internalised by *Coscinodiscus* within 20 minutes, indicative of extensive recycling of the plasma membrane (ref 33, Kuhn and Brownlee 2005, Bot Mar). The extent of dye internalization is not clear from the images presented in Fig 2, but there could be significant dye internalization and redistribution within the 4h period. This could lead to significant labelling of the membranes of the SDV, which may contribute to the labelling patterns observed in Fig 2. Did the authors observe any substantial dye internalization, similar to that observed in *Coscinodiscus*? If so, this should be clearly explained in the manuscript.

A significant difference between the study in *Coscinodiscus* and the one presented here, is that in *Coscinodiscus* FM1-43 was added as a pulse to label the plasma membrane for 5 minutes and then removed, whereas in *S. turris* FM4-64 remained in the media throughout. Dye in the media would have the effect of continuously labelling the plasma membrane, but might also act to obscure dye internalization to some extent. The authors show quite nicely that the extent of FM4-64 labelling increases following exocytosis, which certainly suggests that the SDV membranes are not fully labelled until exposed to the external media, but does not rule out that the SDV membranes are already labelled to some extent by internalized dye. Did the authors also try pulse labelling FM4-64? I think this would help clarify the rate and the extent of dye internalization. As FM dyes are commonly used

as markers of membrane turnover, this approach may also provide further insight into the fate of the plasma membrane following exocytosis, i.e. is the plasma membrane extensively labelled prior to valve exocytosis, we would expect this dye to be rapidly internalized if the plasma membrane is recycled, but the dye will be lost if the plasma membrane disintegrates. This experiment may be difficult to achieve if there is already extensive dye internalization.

In summary, the author should clarify whether FM4-64 is internalized and whether they believe the images shown on Fig 2 primarily represent the plasma membrane, or could also include labelling of the SDV membranes. The statement in line 104 that 'linking extensions are exclusively stained by the membrane dye, indicating that the SDV elongation precedes silicification' seems to suggest that the authors believe that the FM4-46 is also staining the SDV membranes, this should be clarified.

3) Visualization of FM4-64. The 3D reconstructions shown in Fig 2 are very striking and certainly help to visualize the distribution of silica and FM4-64 across the cell as a whole. However, it is difficult to resolve the events at the plasma membrane in this view, and I found the individual Z slices shown in Fig S5 to be much more helpful. In Fig S5 two membranes can be observed at $t=54$, whereas only one can be resolved at $t=48$. Are the authors able to provide more images from the intervening period? Was this cell also labelled with PDMPO, as it would be really useful to visualise the position of the silica during valve exocytosis? It's not clear whether the two membranes can only be resolved at $t=54$ because the position of the plasma membrane has moved away from the valve (as shown in the TEM Fig 3E), or simply because dye can now access the underlying SDV membranes. Is it possible to superimpose the images at $t=48$ and $t=54$ to give an indication of how the position of the plasma membrane has changed?

IN summary, it would be helpful to include some single plane Z images in the main figure 2, ideally with PDMPO included. The single Z sections will also help to demonstrate the extent to which FM4-64 is internalised.

Minor points

Line 38 I found this sentence confusing to read. Rephrase this sentence to 'SDVs are very thin elongated organelles..'?

Line 98 It seems strange to describe FM4-64 as membrane impermeable, as its primary purpose is to integrate into lipid bilayers. Maybe this sentence should read 'FM4-64 is an amphiphilic fluorescent dye that integrates into the plasma membrane, and only crosses the plasma membrane if it is internalized via endocytic vesicles'.

Line 132 The presence of stained membrane remnants on the exocytosed valve was also observed in *Coscinodiscus* cells stained with FM1-43 (ref 33, Kuhn and Brownlee, Bot Mar). It would be nice to cite this observation here.

Line 137 'transiently attached' sounds strange to me. FM dyes are essentially non-fluorescent in aqueous solution but become fluorescent in lipid environment. As the membrane disintegrates, we would expect any FM dyes integrated in the membrane to become non-fluorescent.

Line 167 SVD should be SDV

Fig 3 has some very important detail, but I found the images to be very small. Could it be rearranged to allow the important details to be viewed more clearly?

Reviewer #3 (Remarks to the Author):

Dear Editor,

As requested i have only judged the technical merit of the cryo-electron microscopy work presented in the manuscript.

The authors have done an excellent job. They have obtained excellent results and these are well presented.

The imaging conditions are appropriate for the task, minimizing risk on damage. All information to repeat the imaging experiments is provided, i would just ask the authors to also mention the FIB acceleration voltage (30kV?).

However, as the authors state they have imaged 59 pairs of daughter cells (page 10, line 208), i would like them to provide access to these data. In line with this: the authors report the results shown in figure 4E are representative for what they observe in 8 pairs of cells (line 224/225). it is therefore important to indicate which of the 59 pairs of cells are referred to here.

The same holds for the statement that the authors acquired hundreds of images from 227 cells (page 7, line 150) in the room temperature EM on freeze substituted samples that they used to construct a timeline.

With this taken care off i fully support publication in Nature Comms.

Response to reviewer 1

Reviewer #1 (Remarks to the Author):

The paper by de Haan et al. reports on the membrane dynamics and exocytosis of the silica-based cell wall in diatoms. The mechanisms underlying the intracellular production of an entire mineral-based cell wall and then exocytosis of a rigid structure has been of interest to diatom biologists for decades and various models of exocytosis have been proposed. de Haan uses state-of-the-art microscopy on two diatom species to provide empirical evidence supporting the model that proposes fusion of the proximal SDV membrane with the plasma membrane and then exocytosis of the cell wall. They propose that the proximal SDV membrane is then remodeled and becomes the new plasma membrane of the daughter cell and distal SDV membrane undergoes disintegration. As the authors state, the novelty of this study is the proposition that half of the silicalemma becomes a plasma membrane and the other half is disintegrated and possibly lost.

I am not a structural biologist so I will leave a review of the methodology and validity of the conclusions to others. I will thus base my review on the assumption that the authors' interpretation of the results is robust. All of the models of how diatoms exocytose a fully formed cell wall require some level of fusion between the silicalemma and plasma membrane. What differs among the models is the manner in which membrane homeostasis is maintained. This ranges from membrane recycling through vesicles to repurposing the membrane as the protective organic layer surrounding the frustule. In this study, de Haan et al. proposes a modified mechanism that involves fusion of the proximal SDV membrane which later becomes the new plasma membrane, followed by exocytosis of the cell wall and subsequent disintegration of the remaining silicalemma. Kotzsch et al. 2017 also provides evidence that the proximal SDV membrane fuses with the plasma membrane and although their data hints at distal silicalemma recycling via endocytotic vesicles, they acknowledge the uncertainty of this latter conclusion based on their limited data. So the novelty and significance de Haan et al. can provide is insight into the fate of the distal silicalemma.

Remodeling of the silicalemma as a plasma membrane is an intriguing idea that opens up a whole slew of question. Is this even possible? How would the cell 'swap out' silicalemma-associated proteins with plasma-membrane associated ones? What is known about the lipid composition of the silicalemma vs the plasma membrane? This is not a criticism but rather an acknowledgement that this work stimulates a number of testable hypotheses (which all good science should do) that would hopefully get us closer to understanding the mechanism of cell division in diatoms. However, I do think this could have been discussed more rather than just stating that 'it is likely that both require unique lipid and protein compositions, which would need to be adjusted after fusions' (Lines 275-276). Just a couple of lines about whether there is precedence for this and some citations would at least give some food for thought.

We thank the reviewer for acknowledging the novelty of our work and the constructive feedback. We fully agree that these results incite interesting questions that are worth discussing, and specifically the repurposing of the SDV membrane.

In line with this comment, the relevant paragraph was expanded to include more information about the known compositions of the membranes and possible ways to modify them.

Lines 279-287: "The proposed scenario suggests that upon exocytosis, the proximal SDV membrane becomes the new plasma membrane. As each of the SDV and plasma membranes carry out specialized, different tasks, it is likely that both require unique lipid and protein

compositions, which would need to be adjusted after fusion. Indeed, it has been shown that *T. pseudonana* cells with SDVs have a slightly altered lipid composition, compared to cells that do not contain SDVs, and several proteins have been shown to be specifically associated with the SDV membrane^{2,39–41}. Possible alterations to the proximal SDV membrane to restore plasma membrane specific lipid composition can be mediated by so-called lipid transfer proteins, lipid carriers that transfer specific lipids between donor and acceptor membranes⁴².”

The second novel idea de Haan et al. proposes is large-scale disintegration of the distal SDV membrane in the extracellular space, essentially being loss to the environment, contrasting previous suggestions that the membrane is recycled through endocytotic mechanisms. As support for their conclusions, de Haan et al. run a simulation on the probability of detecting endocytosis. Based on these simulations, they say they would expect to detect 15 endocytotic events out of 68 observations and since they only observed 7, they conclude that membrane internalization cannot be significant. (Lines 270-272). This calculation is somewhat critical to rule out the possibility of membrane recycling as they indicate on Lines 254-258 and support their conclusion of membrane disintegration, but there are some aspects of it that are unclear. First, the surface area of the mature SDV is calculated as ~4000 μm^2 which they state is equivalent to the surface area of the plasma membrane. First, I don't understand why these are equivalent. One daughter cell SDV is not the same size as the entire cell.

We realize that the description of the statistical analysis was insufficient. In the revised version, we added a Supporting Methods section in the supplementary information that significantly expands the methodological section describing how we constructed the statistical model. This will help the reader in understanding more easily what was done and how it supports the conclusions. Regarding the question of the SDV membrane surface area, in *S. turris* the newly formed valve is covering half of the surface of each daughter cell (the other half is covered by the old valve since together they cover the entire cell). Since the SDV membrane completely encloses the valve, on its distal and proximal sides, the surface area of each side of the SDV membrane equals to half the cell's surface area and the two of them together are similar to the surface area of the entire cell. This geometry is now schematically shown in the new Fig. S9, and the reader is referred to this extended methods section in line 264 of the main text.

Text added to the caption of Figure S9: “Note that the length (in this 2D schematic) of the two sides of the SDV membrane is similar to the length of the entire plasma membrane.”

Where does $a = 38$ come from? This should be the side length (incidentally the authors should define these in the methods so those of us who have forgotten geometry don't have to look up the formula) but they state that the cell is 60 μm long. Even if I assume they have taken into account there being 2 daughter cells, I still don't see where 38 comes from.

This is another point that was ill-defined. A full geometrical description with the relevant formulas are now available in the supplementary methods section. Specifically, a capsule is a cylinder with hemispheres on either end. The surface area of a capsule is defined as the sum of the surface area of the cylinder in the center and the combined surface area of the two hemispheres (see figure S9). To calculate the capsule surface area one needs the radius (r) of the hemispheres and the length of the central cylinder (a). *S. turris* cells are 20 to 25 μm in diameter, in the calculation we chose a radius of 11 μm . The length of the cylinder (a) is equal to the total length of the cell minus the radii of the two hemispheres ($60 \mu\text{m} - 11 \mu\text{m} - 11 \mu\text{m} = 38 \mu\text{m}$).

Furthermore, the authors conclude that it is the distal SDV membrane that is being disintegrated so shouldn't their simulation reflect that? If they calculate 15 out of 68 observations for a full SDV, does that mean it would be 7.5 out of 68 for half the SDV? In which case, can they confidently rule out membrane recycling if they indeed observed 7 cases of membrane invagination?

In the scenario we consider in the statistical analysis, the endocytic vesicles should completely recycle the excess membranes after exocytosis. These can be either the whole SDV membrane or the distal SDV membrane plus the old plasma membrane adjacent to it. In any of these cases, recycling of only the distal SDV membrane cannot reflect a complete exocytosis of the new valve. Importantly, the statistical simulation is giving a distribution of possible observations, in which 15 observations was the most likely (but represent only ~20% of the simulated scenarios). The number of simulations in which 10 or less such events were observed is less than 1% (see Fig. S10 A). In addition, the one parameter that is used in the simulation and is fully speculative is the time needed to form and recycle an invagination. The value of 10 seconds that we chose represents an extremely high endocytic reaction rate. In order to show how more conservative values will make the classical endocytosis scenario even less likely, we added to Fig. S10 B an additional simulation in which the invagination time was changed to 20 seconds. In such a scenario we should have seen traces for endocytic events in 15-40 of the cases, indicating an even lower probability for such a scenario.

I think this is overall a very nice study with high quality and modern microscopic approaches and does provide additional support for the mechanism of cell wall exocytosis in diatoms. However, the claim in the title seems to depend heavily on a statistical simulation that, at first pass, does not seem robust and should be clarified.

We hope that the modifications and added information are now sufficient to understand the statistical analysis, and further supports the exocytosis mechanism observed in our experiments.

Response to reviewer 2

Reviewer #2 (Remarks to the Author):

This manuscript describes the process of frustule formation in diatoms, specifically the exocytosis of the mature valve, a remarkable exocytosis event given the size of the newly-formed valve relative to the cell. They demonstrate that the plasma membrane and distal SDV membrane are lost, and the proximal SDV membrane becomes continuous with the plasma membrane. Overall, the study is conducted to a very high degree. The resolution of the imaging, particularly the cryo-ET, is exceptional and provides excellent detail of the process of valve exocytosis. Overall, I find that the findings strongly support the proposed mechanism. The study is of broad interest, not only to the study of biomineralisation but also to cell biologists more generally. I only have a few technical points relating primarily to the live cell imaging that need to be addressed prior to publication.

We thank the reviewer for their interest in our study and for pointing out these important questions regarding membrane dye internalization.

Specific comments

1) Distribution of FM4-64. The live cell imaging of frustule formation and exocytosis used FM4-64 to stain the plasma membrane (Fig 2). Imaging took place over several hours (up to 4 hours). FM4-64 was included in the media throughout the imaging period. The 3D images presented are beautiful and demonstrate that FM4-64 is concentrated in the polygonal pattern seen for PDMPO. I initially found this quite confusing as you might expect the plasma membrane to simply form a smooth continuous layer above the developing SDV. However, the arrangement of the plasma membrane over the edge of the polygonal silica structures can be clearly seen in Fig3C, and this may explain why FM4-64 exhibits this distribution. The authors do say that this is because of ‘close proximity of the plasma membrane to the SDV’ (line 101) but I feel this needs more explanation.

We have modified the relevant text to bring up earlier the fact the the plasma membrane is indeed lining the shape of the SDV.

Lines 101-104: “During silicification, the PDMPO and FM4-64 signals are almost co-localized down to the resolution of the confocal microscope (Fig. 2 A). This demonstrates a very close proximity between the plasma membrane and the growing silica structures within the SDV, namely that the plasma membrane is lining the SDV shape.”

2) Internalization of FM4-64. Previous studies using FM dyes in diatoms have shown that they are rapidly internalized e.g. FM1-43 was rapidly internalised by *Coscinodiscus* within 20 minutes, indicative of extensive recycling of the plasma membrane (ref 33, Kuhn and Brownlee 2005, Bot Mar). The extent of dye internalization is not clear from the images presented in Fig 2, but there could be significant dye internalization and redistribution within the 4h period. This could lead to significant labelling of the membranes of the SDV, which may contribute to the labelling patterns observed in Fig 2. Did the authors observe any substantial dye internalization, similar to that observed in *Coscinodiscus*? If so, this should be clearly explained in the manuscript.

In our experiments there was no indication for internalization of the FM4-64 dye. We have conducted additional experiments with the aim of investigating any possible differences between FM4-64 and FM1-43. The results show considerable differences, namely that FM1-43 is indeed internalized also in *S. turris* (similar to the results indicated by the reviewer), but FM4-64 is not substantially internalized. These results are now presented in the new Fig. S4. These data corroborate our conclusions that the labelling patterns in Figure 2 are the result of plasma membrane staining exclusively.

A significant difference between the study in *Coscinodiscus* and the one presented here, is that in *Coscinodiscus* FM1-43 was added as a pulse to label the plasma membrane for 5 minutes and then removed, whereas in *S. turris* FM4-64 remained in the media throughout. Dye in the media would have the effect of continuously labelling the plasma membrane, but might also act to obscure dye internalization to some extent. The authors show quite nicely that the extent of FM4-64 labelling increases following exocytosis, which certainly suggests that the SDV membranes are not fully labelled until exposed to the external media, but does not rule out that the SDV membranes are already labelled to some extent by internalized dye. Did the authors also try pulse labelling FM4-64? I think this would help clarify the rate and the extent of dye internalization. As FM dyes are commonly used as markers of membrane turnover, this approach may also provide further insight into the fate of the plasma membrane following exocytosis, i.e. is the plasma membrane extensively labelled prior to valve exocytosis, we would expect this dye to be rapidly internalized if the plasma membrane is recycled, but the dye will be lost if the plasma membrane

disintegrates. This experiment may be difficult to achieve if there is already extensive dye internalization.

We performed the suggested pulse experiments, and the results are now part of Fig. S4. The data show that FM1-43 is indeed internalized in *S. turris* and we can detect its fluorescence in intracellular membranes (such as in chloroplasts and the nucleus) after a short pulse. On the other hand, no FM4-64 labelling remains after a pulse stain, indicating that the FM4-64 interacts more loosely with the plasma membrane and if washed from the medium it is released from the membrane and the staining is lost. This strengthens our conclusions that the staining we show in Figure 2 is exclusively due to continuous staining of the plasma membrane.

In summary, the author should clarify whether FM4-64 is internalized and whether they believe the images shown on Fig 2 primarily represent the plasma membrane, or could also include labelling of the SDV membranes. The statement in line 104 that 'linking extensions are exclusively stained by the membrane dye, indicating that the SDV elongation precedes silicification' seems to suggest that the authors believe that the FM4-46 is also staining the SDV membranes, this should be clarified.

The additional results now show that FM4-64 staining is not internalized in any significant amount (demonstrated by the complete loss of the stain after washing), thus the images in Figure 2 indeed represent the plasma membrane.

Regarding the sentence about the staining at the tips of the linking extensions, also here we suggest exclusive staining of the plasma membrane, that indeed co-localizes with the SDV membrane in the confocal microscope. The point we intended to make is that we see the plasma membrane elongated further than the silica (there is a gap between silica labeling and membrane labeling), suggesting that at the tip of the extending SDV there is some volume of the SDV lumen that will subsequently get filled with silica. The phrasing was revised to make the point clearer: Lines 105-107: "Nevertheless, the growing tips of the linking extensions are stained only with the membrane dye, indicating that SDV elongation precedes the silicification of its most marginal parts (Fig. 2 A, arrow)."

3) Visualization of FM4-64. The 3D reconstructions shown in Fig 2 are very striking and certainly help to visualize the distribution of silica and FM4-64 across the cell as a whole. However, it is difficult to resolve the events at the plasma membrane in this view, and I found the individual Z slices shown in Fig S5 to be much more helpful. In Fig S5 two membranes can be observed at t=54, whereas only one can be resolved at t=48. Are the authors able to provide more images from the intervening period?

The temporal resolution of the time lapses was one z-stack every 6 minutes, this in an experimental limitation imposed by laser-induced stress to the cells, therefore we do not have images from the intervening period.

Was this cell also labelled with PDMPO, as it would be really useful to visualise the position of the silica during valve exocytosis?

The required excitation wavelength for the PDMPO stain (405 nm) was quickly phototoxic to diatom cells. We were able to record PDMPO and FM4-64 along several (3 to 5) snap shots with low temporal resolution (every 20 to 60 minutes), without causing significant stress. However, in

order to record the detailed time lapses we had to omit the PDMPO imaging. Since the snap shots allowed us to establish the very close resemblance between the PDMPO and FM4-64 dynamics, the FM4-64 staining pattern by itself was enough to infer the silica formation stage in subsequent time lapses.

Nevertheless, in order to provide more information to the reader of the various staining experiments we added an additional dataset to the new Figure S6 (old Figure S5). This dataset includes both PDMPO and FM4-64 staining, but only two time points. These images present z-slices that demonstrate how the two signals overlap before exocytosis but become different close to exocytosis as the plasma membrane does no longer follow the exact silica features.

It's not clear whether the two membranes can only be resolved at t=54 because the position of the plasma membrane has moved away from the valve (as shown in the TEM Fig 3E), or simply because dye can now access the underlying SDV membranes. Is it possible to superimpose the images at t=48 and t=54 to give an indication of how the position of the plasma membrane has changed?

In cells with intact SDVs, the plasma membrane, SDV membrane and silica are too close to each other (<150 nm) to distinguish them from each other. We believe that the fact that we can resolve two separate membranes at time point t=54 indicates that these membranes have moved away from each other, which corroborates our TEM observations of *S. turris* (Fig. 3). Having said the former, we tried to overlay the images of the two time points, but the overall resolution is not sufficient to resolve if indeed there is any membrane movement (see below this unfruitful attempt).

IN summary, it would be helpful to include some single plane Z images in the main figure 2, ideally with PDMPO included. The single Z sections will also help to demonstrate the extent to which FM4-64 is internalised.

We have included such additional data in Figure S6 as explained above.

Minor points

Line 38 I found this sentence confusing to read. Rephrase this sentence to 'SDVs are very thin elongated organelles..'

we revised the sentence as suggested.

Lines 38-39: "SDVs are thin and elongated organelles, positioned under the plasma membrane."

Line 98 It seems strange to describe FM4-64 as membrane impermeable, as it is primary purpose is to integrate into lipid bilyaers. Maybe this sentence should read 'FM4-64 is an amphiphilic

fluorescent dye that integrates into the plasma membrane, and only cross the the plasma membrane if it is internalized via endocytic vesicles’.

Thank you for this suggestion, we modified the text according to this suggestion and also to include the added information about the FM1-43.

Lines 98-101: “FM4-64 is an amphiphilic fluorescent dye that integrates into the plasma membrane³². Unlike similar dyes, such as FM1-43 that is rapidly internalized in a different diatom³³, FM4-64 labels only the plasma membrane and does not passively infiltrate into *S. turris* cells (Fig. S4).”

Line 132 The presence of stained membrane remnants on the exocytosed valve was also observed in *Coscinodiscus* cells stained with FM1-43 (ref 33, Kuhn and Brownlee, Bot Mar). It would be nice to cite this observation here.

Thank you for this suggestion, we added the reference to this part of the text.

Lines 136-137: “This is in accordance with previously reported membrane fluorescence study of *Coscinodiscus wailesii*³³.”

Line 137 ‘transiently attached’ sounds strange to me. FM dyes are essentially non-fluorescent in aqueous solution but become fluorescent in lipid environment. As the membrane disintegrates, we would expect any FM dyes integrated in the membrane to become non-fluorescent.

We revised the text so accordingly.

Lines 141-142: “ It is important to note that FM4-64 is present in the medium throughout the experiment but only becomes fluorescent in a lipid environment.”

Line 167 SVD should be SDV

We thank the reviewer for noticing this typo, we corrected it. Line 172.

Fig 3 has some very important detail, but I found the images to be very small. Could it be rearranged to allow the important details to be viewed more clearly?

We rearranged the panels of Figure 3 in a vertical arrangement. The actual size of each panel is now 50% larger than in the original figure.

Response to reviewer 3

Dear Editor,

As requested i have only judged the technical merit of the cryo-electron microscopy work presented in the manuscript.

The authors have done an excellent job. They have obtained excellent results and these are well presented.

The imaging conditions are appropriate for the task, minimizing risk on damage. All information to repeat the imaging experiments is provided, i would just ask the authors to also mention the FIB acceleration voltage (30kV?).

We thank the reviewer for their positive feedback and noticing this missing parameter, we have added it now.

Lines 399-400: "The lamellae were milled at a 12° tilt relative to the grid plane with the rough milling (to 1 μm thickness) involving 2 steps using the gallium ion beam at 30 kV and a current of 750 pA and 300 pA."

However, as the authors state they have imaged 59 pairs of daughter cells (page 10, line 208), I would like them to provide access to these data. In line with this: the authors report the results shown in figure 4E are representative for what they observe in 8 pairs of cells (line 224/225). It is therefore important to indicate which of the 59 pairs of cells are referred to here.

Out of the 59 pairs of cells, 15 were close to the exocytosis event and were analyzed in the work (as stated in the main text). In accordance with this comment, in the Readme file that is part of the Dryad deposition, we indicate how these 15 cells (containing 26 datasets since for some cells we have more than one dataset) are divided to early exocytosis, recent exocytosis, and late exocytosis.

The same holds for the statement that the authors acquired hundreds of images from 227 cells (page 7, line 150) in the room temperature EM on freeze substituted samples that they used to construct a timeline.

We have uploaded all the tomograms and room temperature TEM images to the Dryad repository. Lines 428-430: "Data availability
All source data generated or analyzed during this study have been deposited to Dryad"

With this taken care of I fully support publication in Nature Comms.

Thank you for the positive feedback.

REVIEWERS' COMMENTS

Reviewer #1 (Remarks to the Author):

I appreciate the additional information the authors have included and feel they have adequately addressed my concerns.

Thank you for a nice contribution to understanding diatom silicification.

Sincerely,

Kim Thamatrakoln

Reviewer #2 (Remarks to the Author):

The authors have addressed all of my concerns, adding substantial amounts of additional data to demonstrate that the FM4-64 dye is not internalised. I have no further concerns about the manuscript.